# Hypergraph reconstruction from dynamics

Robin Delabays [1], Giulia De Pasquale[2], Florian Dörfler[3] & Yuanzhao Zhang [4] ✉

A plethora of methods have been developed in the past two decades to infer the underlying network structure of an interconnected system from its collective dynamics. However, methods capable of inferring nonpairwise interactions are only starting to appear. Here, we develop an inference algorithm based on sparse identification of nonlinear dynamics (SINDy) to reconstruct hypergraphs and simplicial complexes from time-series data. Our model-free method does not require information about node dynamics or coupling functions, making it applicable to complex systems that do not have a reliable mathematical description. We first benchmark the new method on synthetic data generated from Kuramoto and Lorenz dynamics. We then use it to infer the effective connectivity in the brain from resting-state EEG data, which reveals significant contributions from non-pairwise interactions in shaping the macroscopic brain dynamics.

Hypergraphs and simplicial complexes have emerged as versatile and powerful tools for modeling and analyzing complex systems, offering flexible and expressive representations of higher-order relationships and dependencies that traditional graphs cannot capture[1-5]. These higher-order structures play an important role in a wide range of dynamical processes, from brain dynamics[6,7] to communications in social systems[8,9] and competitions in ecological systems[10,11].

Broadly speaking, the analysis of coupled dynamical systems can go in two directions. On the one hand, it is important to predict the possible dynamics based on model equations and coupling structures. Many recent studies have taken this approach and investigated how higher-order interactions can influence collective dynamics[12-19] such as diffusion[20], consensus[21,22], contagion[23,24], synchronization[25-29], and controllability[30]. On the other hand, the inverse problem is equally important. Namely, is it possible to recover the coupling structure from dynamics? This inference problem is important in many fields, including neuroscience[31-34], e.g., infer effective connectivity between brain regions from functional Magnetic Resonance Imaging (fMRI) data, and epidemiology[35,36], e.g., reconstruct social interactions from COVID infection data. In neuroscience, for example, hypergraph inference techniques have been shown to help identify biomarkers in the autism spectrum disorder, potentially suggesting novel therapeutic interventions for such disorders[37].

When considering only pairwise interactions, network inference is a relatively mature field[38-49] and has been approached from numerous perspectives utilizing diverse tools such as sparse regression[50], Bayesian inference[51], reservoir computing[52], causation entropy[53], transfer entropy[54], and Granger causality[55]. In comparison, the study of hypergraph inference is still in its infancy[4]. Luckily, progress is happening rapidly[56-60]. Some recently proposed methods include probabilistic inference based on prior network data[61-63] or binary contagion data[64] and optimization-based methods that apply to time-series data[65]. These methods, however, are limited by their strong assumptions of the generative process and/or reliance on knowledge of the model dynamics. When the underlying model is unknown, the literature on hypergraph inference is rather scarce. To the best of our knowledge, the only model-free method for the causal inference of higher-order interactions is the Algorithm for Revealing Network Interactions (ARNI)[44].

Here, we propose the Taylor-based Hypergraph Inference using SINDy (THIS) algorithm, which reconstructs hypergraphs and simplicial complexes from time-series data. Our method is system-agnostic, noninvasive, and distributed—it does not require knowledge of the node dynamics or coupling functions and does not require curating different nonlinear feature libraries for each application. Neither does it require perturbing the system in precise ways (e.g., via control input injection). Moreover, the inference can be made for each node independently, making the computation easily parallelizable. Importantly, the method takes advantage of the intrinsic sparsity of most real-world hypergraphs through the use of the SINDy algorithm[66]. We apply THIS

[1]School of Engineering, University of Applied Sciences of Western Switzerland HES-SO, Sion, Switzerland. [2]Department of Electrical Engineering, Eindhoven University of Technology, Eindhoven, The Netherlands. [3]Department of Information Technology and Electrical Engineering, ETH Zürich, Zürich, Switzerland. [4]Santa Fe Institute, Santa Fe, NM, USA. ✉e-mail: yzhang@santafe.edu

to both synthetic data generated from canonical dynamical systems and resting-state EEG data from 109 human subjects. With the synthetic data, we show that THIS can be extremely data efficient and compares favorably against ARNI in accuracy. From the EEG data, we find that higher-order interactions play a key role in shaping macroscopic brain dynamics (despite most physical connections between brain regions being pairwise[33]).

## Results

### When is hypergraph reconstruction possible?

We consider $n$ nonlinear systems coupled through a hypergraph, as described by the following equations:

$$\dot{x}_i = F_i(\mathbf{x}) = f_i(x_i) + \sum_{j=1}^{n} a_{ij}^{(2)} h^{(2)}(x_i, x_j)$$
$$+ \sum_{j,k=1}^{n} a_{ijk}^{(3)} h^{(3)}(x_i, x_j, x_k) + \cdots, \quad i = 1, \cdots, n, \quad (1)$$

where $f_i$ describes the intrinsic dynamics of node $i$. The adjacency tensor $A^{(p)} = \{a_{ijk\ldots\ell}^{(p)}\}$ determines which nodes are coupled through the $p$-th order interaction function $h^{(p)}$.

Before developing an inference method, it is helpful to characterize when is inference theoretically possible. For this purpose, we assume that it is possible to observe all nodes in the system for many different initial conditions at high resolution without the interference of noise. In this perfect and abundant data limit, the question becomes: Can we determine the hypergraph structure uniquely from the vector field? When will there be ambiguity? Below, for simplicity and without loss of generality, we consider systems with interactions up to the third order (i.e., $A^{(p)} = \mathbf{0}$ for $p > 3$).

Neuhäuser et al.[21] recently pointed out that, for linear consensus dynamics, pairwise and higher-order interactions are theoretically indistinguishable from each other. Here, we generalize this observation to any dynamical systems on hypergraphs (similar arguments can also be found in ref. [67]). If the triadic interaction function $h^{(3)}(x_i, x_j, x_k)$ can be written as a linear combination of pairwise interaction functions $h^{(3)}(x_i, x_j, x_k) = h'(x_i, x_j) + h''(x_i, x_k) + h'''(x_j, x_k)$ for all $\mathbf{x}$, then there is no formal difference between a 2-simplex and a closed triangle of links. In other words, for any hypergraph with linearly decomposable interaction functions, we can always find a corresponding network that generates the same dynamics. As a consequence, hypergraph reconstruction would be impossible due to such ambiguity. More generally, if we only have data from a localized region of the state space (e.g., by observing the system respond to small noise around a fixed point), inside which the dynamics can be effectively linearized, then it is not possible to reconstruct the hypergraph from dynamics even when the interactions are nonlinear.

On the other hand, if a higher-order interaction cannot be decomposed into a linear combination of pairwise interactions, then it will produce a vector field that is different from any vector field generated with only pairwise interactions. This difference can, in principle, be used to infer the existence of higher-order interactions. Next, we propose a simple strategy to extract this information, which can then be used to reconstruct the causal hypergraph.

### Taylor-based Hypergraph Inference using SINDy (THIS)

In equation (1), one can write the Taylor expansion of the dynamics $F_i$ around an arbitrary point $\mathbf{x}_0$, leading to

$$\dot{x}_i \approx F_i(\mathbf{x}_0) + \sum_j \partial_j F_i(\mathbf{x}_0)\Delta x_j + \frac{1}{2!}\sum_{j,k} \partial_{j,k} F_i(\mathbf{x}_0)\Delta x_j \Delta x_k + \cdots \quad (2)$$

with $\Delta x_i = x_i - x_{0,i}, \forall i \in \{1, \ldots, n\}$. One realizes that, for $i \neq j \neq k$, if the coefficient $\partial_{j,k} F_i(\mathbf{x}_0)$ is nonzero, then there is necessarily a triadic

interaction involving nodes $i$, $j$, and $k$ (more precisely, nodes $j$ and $k$ would influence node $i$ jointly through a directed triadic interaction). One can see this by noticing that $\partial_{j,k} h^{(2)}(x_i, x_j) = 0$ for $i \neq k$, whereas $\partial_{j,k} h^{(3)}(x_i, x_j, x_k)$ is generally nonzero. Our approach utilizes this observation and aims to infer the nonzero coefficients $\partial_{j,k} F_i$ from time series. Note that since we don't know the form of the coupling functions, there is no point in inferring the weights of the hyperedges, so we focus on the binary inference problem. Namely, is there a hyperedge or not?

Importantly, this Taylor-based approach is intrinsically system-agnostic and does not rely on knowing the node dynamics $f_i$ or coupling functions $h^{(p)}$. Moreover, the Taylor expansion can be computed around any point $\mathbf{x}_0$ where the vector field is differentiable, rendering the approach flexible in terms of where data are collected. In rare circumstances (usually with zero probability), a Taylor coefficient could vanish when evaluated at $\mathbf{x}_0$ even when the corresponding interaction exists. We can easily circumvent this problem by choosing a different base point $\mathbf{x}_0$.

To recover the coefficients of the Taylor expansion based on a time series $\mathbf{x}(t)$, we use SINDy[66] with a library of monomials up to a chosen degree (see Methods section for details). The purpose of SINDy is to find a parsimonious linear combination of the library functions that best explain the time-series data $X = \{\mathbf{x}(t_1), \mathbf{x}(t_2), \cdots, \mathbf{x}(t_K)\}$. Each nonzero coefficient obtained by SINDy selects a monomial from the library, which can in turn be used to infer the corresponding hyperedge (note that a monomial of degree $p - 1$ in the Taylor expansion corresponds to an interaction of order $p$). Specifically, we consider a hyperedge to exist if the coefficient of the corresponding monomial is above a prespecified threshold $\epsilon$. The threshold $\epsilon$ has overlapping functionalities with the sparsity parameter in SINDy. Having such a separate threshold facilitates measuring the quality of the inference – it allows us to draw the Receiver Operating Characteristic (ROC) curves a posteriori by adjusting $\epsilon$ without having to rerun the sparse regression algorithm multiple times. We illustrate our inference procedure in Fig. 1.

Our main rationale for using SINDy is that many real-world hypergraphs are intrinsically sparse. As each nonzero coefficient corresponds to a hyperedge, an algorithm promoting sparsity is desirable in most cases. Of course, dense hypergraphs do exist. However, in such cases, the actual hypergraph structure has only marginal importance, as most phenomena on dense hypergraphs are well captured by mean-field approximations. On top of that, there are two additional advantages of using SINDy. First, we can naturally control the trade-off between sensitivity and specificity of the reconstruction by tuning the sparsity parameter in SINDy. Second, the SINDy-based approach also allows us to use mature, off-the-shelf implementations of the algorithm that are user-friendly and highly optimized[68]. For these reasons, we focus on the SINDy implementation of our approach in this paper. However, it is possible to combine our Taylor-based approach with other data-driven methods, such as Extended Dynamic Mode Decomposition (EDMD)[69].

We would like to emphasize that our inference procedure can be done independently for each node. Thus, if one is only interested in a subset of nodes, there is no need to infer the full hypergraph. It also means that the more nodes there are, the more parallelizable our method will be, which is important for dealing with high-dimensional time series from large interconnected systems.

Finally, we remark that THIS has a few similarities with ARNI[44]. Indeed, the basic idea of both approaches is to represent the unknown coupling functions as linear combinations of a set of nonlinear basis functions. Namely, monomials for THIS and a user-specified function library for ARNI. However, the ways in which THIS and ARNI identify the coefficients of the decomposition (i.e., the optimization strategies) are fundamentally different. For ARNI, the coefficients are determined by consecutively projecting the time series onto the subspaces

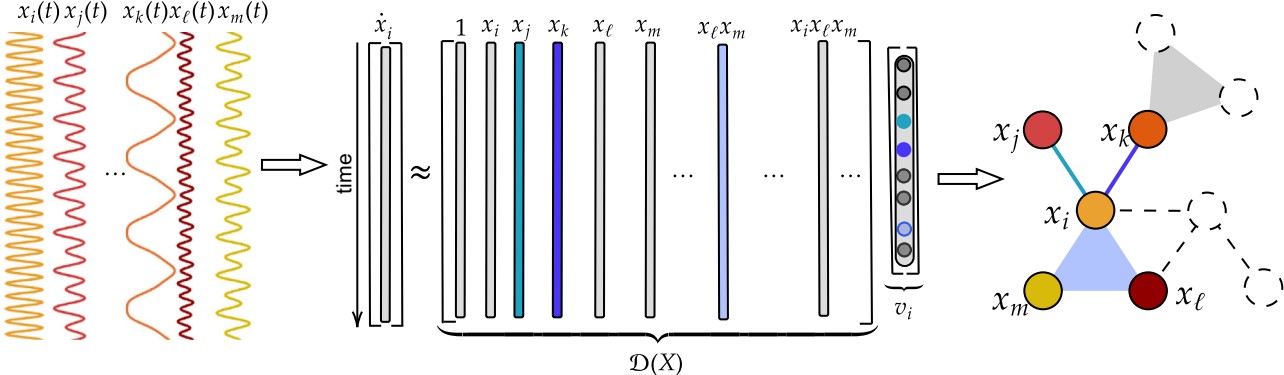

**Fig. 1 | Illustration of Taylor-based Hypergraph Inference using SINDy (THIS).** Here, we focus on inferring the couplings received by node $i$ (highlighted in color on the right). Inference for any other node can be done independently using the same procedure. For each inference task, the input is the time series measured for all nodes (left), and the output is the inferred connections pointing towards node $i$ (right). The key step of the inference is solving the matrix equation $\dot{\mathbf{x}}_i = \mathcal{D}(X)\mathbf{v}_i$ (middle). On the left-hand side, $\dot{\mathbf{x}}_i = [\dot{x}_i(t_1), \dot{x}_i(t_2), \cdots, \dot{x}_i(t_K)]^\top$ is a column vector consisting of the derivative of $x_i$ at different time points. On the right-hand side, $\mathcal{D}(X)$ is the data matrix obtained by applying nonlinear features (i.e., the monomials) to the time series, and $\mathbf{v}_i$ is a sparse vector that approximately solves the matrix equation. Note that, to be precise, the variables $\{x_i\}$ in $\mathcal{D}(X)$ should be interpreted as deviations from the Taylor-expansion base point $\mathbf{x}_0$. The nonzero elements of $\mathbf{v}_i$ (highlighted with colors) are used to infer the existence of (hyper) edges. In this example, there is a triadic coupling from nodes $\ell$ and $m$ to node $i$ because the coefficient for $x_\ell x_m$ is nonzero.

spanned by the basis functions. THIS, on the other hand, performs inference through a sparsity-promoting optimization procedure. Next, we compare the performance of the two algorithms under identical conditions using synthetic data.

## Benchmarks on synthetic data

Before applying the new inference method to real-world data, it is important to benchmark it on synthetic data, for which ground truth is available to validate the inference. We benchmark THIS on synthetic data generated from two dynamical systems: a higher-order Kuramoto model and Lorenz oscillators coupled through nonpairwise interactions.

**Higher-order Kuramoto model.** We first compare the inference performance of ARNI and THIS using a generalization of the Kuramoto model:

$$\dot{\theta}_i = \omega_i + \sum_{j=1}^{n} a_{ij}^{(2)} \sin(\theta_j - \theta_i) + \sum_{j,k=1}^{n} a_{ijk}^{(3)} \sin(\theta_j + \theta_k - 2\theta_i), \quad i = 1, \ldots, n, \quad (3)$$

where $\theta_i \in S^1$ represents the phase of oscillator $i$ and $\omega_i$ is its natural frequency. As a proof of concept, we start the test with the seven-node hypergraph shown in Fig. 2. (see Supplementary Fig. 1 for a comparison between ARNI and THIS on pairwise networks). Samples are taken randomly and uniformly inside a hypercube of side length $\delta = 0.1$ centered at the origin $\mathbf{x}_0 = \mathbf{0}$. One can use hypercubes of side lengths up to $\delta = 1.0$ without affecting the results, see Supplementary Fig. 2. For each data point, we compute the derivatives directly using Eq. (3).

The output of both inference methods (THIS and ARNI with power-series basis) is a weighted adjacency tensor $A^{(p)}$ for each interaction order $p$. Then one needs to choose a threshold $\epsilon$ such that a hyperedge is considered present if and only if the corresponding coefficient in $A^{(p)}$ is greater than $\epsilon$. For a given $\epsilon$, the true positive rate (TPR) [resp. false positive rate (FPR)] is the percentage of existing [resp. nonexistent] hyperedges that were inferred. The ROC curves plot TPR against FPR for all possible threshold values.

In Fig. 2, we show the ROC curves for THIS and ARNI applied to samples of varying sizes. Among the different ROC curves, we vary the sample size from 10 to 150 (curves with brighter colors use more data points). Along each ROC curve, we gradually decrease the inference threshold $\epsilon$ from $\infty$ to 0. Thus, each curve starts at the lower left corner (0, 0), where no (hyper)edge is inferred, and ends at the upper right

corner (1, 1), where all possible (hyper)edges are inferred. Since we want high TPR and low FPR, an ideal ROC curve should rise steeply from the lower left corner to the upper left corner. A diagonal curve is indicative of performance comparable to random guesses. We see that even with just 10 data points, THIS achieves good accuracy in reconstructing the hypergraph. In contrast, ARNI barely outperforms random guesses even with 150 data points.

Our algorithm can handle larger systems as well. Figure 3 and Supplementary Fig. 5 show that THIS can accurately infer a 100-node random simplicial complex from generalized Kuramoto dynamics. We also show similar results for large random hypergraphs in Supplementary Fig. 6.

**Lorenz oscillators with nonpairwise coupling.** Next, we apply THIS to Lorenz oscillators coupled through pairwise and triadic interactions:

$$\dot{x}_i = \sigma(y_i - x_i) + \sum_{j=1}^{n} a_{ij}^{(2)}(x_j - x_i) + \sum_{j,k=1}^{n} a_{ijk}^{(3)}(x_j x_k^2 - x_i^3),$$
$$\dot{y}_i = x_i(\rho - z_i) - y_i,$$
$$\dot{z}_i = x_i y_i - \beta z_i,$$

where we set $\sigma = 10$, $\rho = 28$, and $\beta = 8/3$. Here, each node has three degrees of freedom. We perform inference on the whole system as if it were a $3n$-node system. Then, we aggregate the cross-node interactions: For each pair of nodes, say $i$ and $j$, we use the largest inferred coefficient between any degree of freedom of node $i$ and any degree of freedom of node $j$ as $a_{ij}^{(2)}$. We perform the same procedure to determine $a_{ijk}^{(3)}$.

In Fig. 4, we show that our method performs well even for chaotic dynamics and nodes with more than one dimension. Each ROC curve corresponds to a different five-node hypergraph generated using XGI's `random_hypergraph` function[70], with a probability of 0.5 for both 2-edges and 3-edges. For each hypergraph, we use 10 independent time series for the inference, each consisting of 150 time steps with a step size $\delta_t = 0.01$, and we approximate the derivatives through finite differences. The time series are all initialized within a hypercube of side length $\delta = 0.7$, centered at a random point $\mathbf{x}_0$ chosen uniformly from $[-1, 1]^{3n}$. For most hypergraphs, we can reach over 80% TPR with less than 20% FTR. The performance can be further improved if we use more data points, optimize hyperparameters such as the sampling box

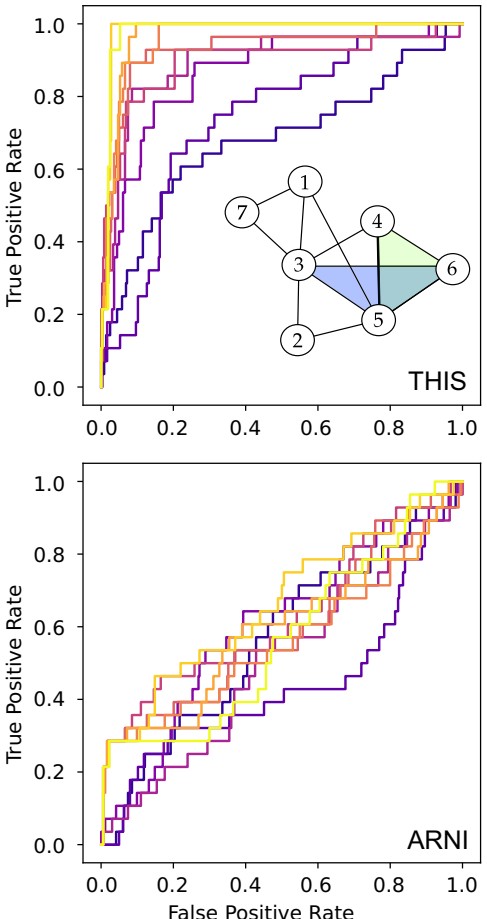

**Fig. 2 | THIS versus ARNI applied to Kuramoto dynamics.** We infer the hypergraph shown in the top panel. The ROC curves measure the inference quality considering all orders of interactions (both pairwise and triadic). Each curve corresponds to a different sample size used in inference, ranging from 10 data points (dark purple) to 150 data points (bright yellow). The quicker the ROC curve rises to the upper left corner, the better the inference. A breakdown of the ROC curves according to interaction orders is available in Supplementary Fig. 4.

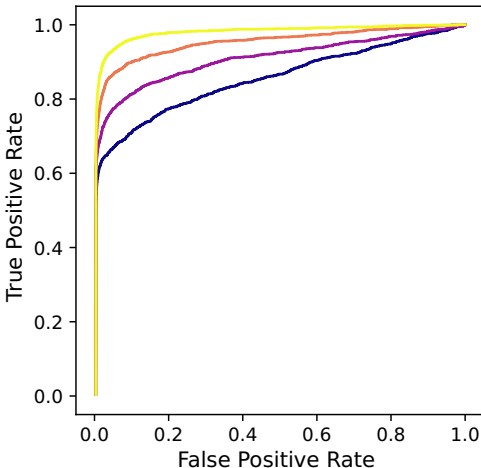

**Fig. 3 | Inference of a 100-node simplicial complex with generalized Kuramoto dynamics.** Each curve corresponds to a different sample size used in inference, ranging from 500 data points (dark purple) to 2000 data points (bright yellow). The simplicial complex was randomly generated through an Erdős-Rényi-type process: each 2-edge (resp. each 3-edge including its boundary) exists with probability 1% (resp. 0.1%).

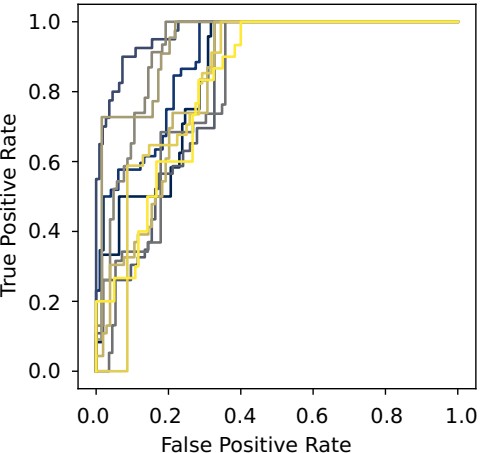

**Fig. 4 | THIS applied to Lorenz oscillators on random hypergraphs.** Each ROC curve corresponds to the inference of a different five-node random hypergraph. To test the robustness of THIS, we made the hypergraphs non-sparse ($p = 0.5$ for both pairwise and triadic connections) and estimated derivatives from numerical differentiation. Since each Lorenz oscillator has three degrees of freedom, we are essentially performing inference on a 15-dimensional system.

size, or calculate the derivatives directly from the underlying Lorenz equations.

We note that for THIS, the existence of higher-order interactions could mask the existence of lower-order interactions. This stems from an unavoidable drawback of relying on the Taylor expansion: Higher-order interactions also contribute to lower-order Taylor coefficients. Indeed, the existence of interactions of order $p$ implies that the corresponding coefficients of order lower than $p$ can be nonzero as well. In other words, when a $p$-edge $e$ is present in the hypergraph, THIS will likely infer some lower-order edges contained in $e$ even if they did not exist, potentially leading to false positives. Despite this limitation, THIS performed well in our benchmark with random hypergraphs (Fig. 4 and Supplementary Fig. 6). We will discuss practical strategies to further mitigate this issue in the Discussion section.

### How important are higher-order interactions in shaping macroscopic brain dynamics?

An important special case of the hypergraph reconstruction problem is the following: Given time-series data, can we tell whether networks are adequate to capture the observed dynamics, or are higher-order interactions truly needed[67,71]? Answering this question has important applications in neuroscience: Although different brain regions are mostly connected through (pairwise) anatomical axons and nerve

fibers, the release of neurotransmitters can potentially induce higher-order interactions among neuronal populations. So how important are nonpairwise interactions in shaping the macroscopic brain dynamics?

Below, we apply THIS to reconstruct the effective connectivity between seven brain regions using resting-state EEG data from 109 human subjects. For each subject, two independent recordings were collected, giving a total of 218 time series (see Methods section for details of the analysis protocol). Interestingly, we find that non-pairwise interactions account for more than 60% of the EEG dynamics, and this is robust across brain regions and individual subjects.

To measure the amount of dynamics that is explained by higher-order interactions we compute the relative contribution of each interaction order $\ell$ to the derivative. Specifically, if $\hat{a}_{i_1,\ldots,i_\ell}^{(\ell)}$ is the

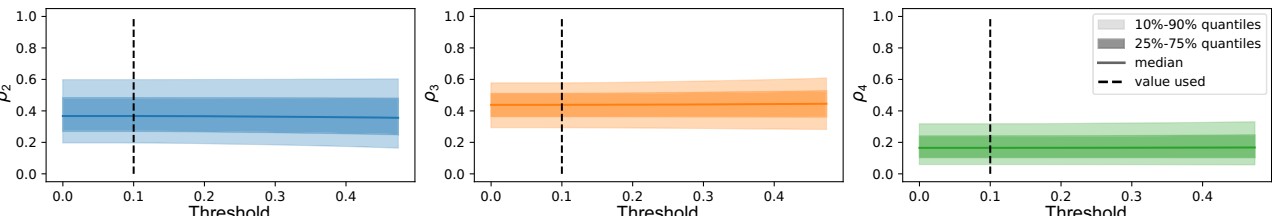

**Fig. 5 | Statistics of each interaction order's contribution to the dynamics for the coarse-grained EEG data.** For each threshold value (deciding whether a hyperedge is inferred or not), we compare the contribution of 2nd-, 3rd-, and 4th-order interactions to the dynamics. We show here the median (plain line), the 25%-75% quantiles (dark area), and the 10%-90% quantiles (light area) of these contributions over the 218 time series considered. Notice that for any threshold value, $\rho_2 + \rho_3 + \rho_4 = 1$. The contribution of each interaction order is fairly consistent across a wide range of threshold values.

inferred adjacency tensor for $\ell$-th order interactions, we compute the $\ell$-th order contribution ratio

$$\rho_\ell = \frac{\sum_{i_1 \neq \ldots \neq i_\ell} |\hat{a}^{(\ell)}_{i_1, \ldots, i_\ell} x_{i_2} \ldots x_{i_\ell}|}{\sum_{k=2}^{o} \sum_{i_1 \neq \ldots \neq i_k} |\hat{a}^{(k)}_{i_1, \ldots, i_k} x_{i_2} \ldots x_{i_k}|}, \qquad (4)$$

where $o$ is the maximal interaction order considered. Following the discussion in the Supplementary Information (section S7), here we take $o = 4$. The ratios $\rho_\ell$ can be computed for each time series $s$ and at each time step $t$, hence we index the ratios $\rho_\ell(s, t)$. As we are interested in the relative contribution from higher-order interactions, we conveniently denote the ratio of their contribution as $\rho(s, t) = \rho_3(s, t) + \rho_4(s, t)$. We then sort the time series according to their median higher-order ratio $\bar{\rho}(s) = \text{median}_t(\rho(s, t))$.

Figure 5 shows the contribution of the second, third and fourth order interactions to the brain dynamics. We see that the second-order and third-order interactions are the ones that contribute the most in shaping the brain dynamics across all inference thresholds. Unlike the synthetic cases from the last section, here we no longer have access to the ground truth, which means that we can no longer rely on ROC curves to pick the optimal threshold value for $\epsilon$. Luckily, Fig. 5 shows that our finding is robust and not sensitive to the choice of the threshold value, since the inferred $\rho_\ell$ stays almost invariant across the entire plausible range of $\epsilon$.

In Fig. 6, we show the distribution of $\rho(s, t)$. The left-most violin plot shows the distribution of $\rho(s, t)$ aggregated over all time series and all time steps, whereas the other violin plots show the distribution for individual time series from different percentiles (ranked by their median ratios). Our inference results are robust and consistent across different subjects: For the majority of the time series, between 60% and 70% of the dynamics are attributed to nonpairwise interactions. For a breakdown of the contribution from each interaction order, see Supplementary Fig. 9. The right panel in Fig. 6 shows the six most prominent 3-edges inferred by THIS. Interestingly, all of them involve the prefrontal cortex, which matches the expectation that the prefrontal cortex acts as a major information processing center in the brain[72]. We confirm that the most prominent 4-edges (not shown in Fig. 6) also overwhelmingly point toward the prefrontal cortex.

Our finding raises the question about the neurological basis of higher-order interactions in the brain. Some potential mechanisms for creating nonpairwise interactions among neuronal populations include heterosynaptic plasticity, ephaptic transmission, extracellular fields, and metabolic regulation[73,74]. Even from purely pairwise structural connectivity, nonpairwise effective connectivity can naturally emerge (e.g., through coarse-graining). It is also worth emphasizing that the goal of THIS is not inferring functional connectivity or structural connectivity, which have been tackled in a model-free fashion previously[75-77]. Instead, we are inferring effective connectivity, in the same spirit of the dynamic causal modeling paradigm[31].

We note that currently there is no consensus on the relative importance of pairwise versus nonpairwise interactions in shaping the macroscopic brain dynamics[76,78-84]. Due to the intrinsic complexities of the brain and a lack of ground truth, different methods applied to different neurophysiological datasets can lead to seemingly contradictory results. For example, a recent work[82] based on intracranial encephalography (iEEG) and fMRI data suggests that linear auto-regressive models provide the best fit for macroscopic brain dynamics in resting-state conditions both in terms of one-step predictive power and computational complexity. Ref. 76, on the other hand, supports the existence of higher-order interactions based on partial entropy decomposition to resting-state fMRI data from human brains. Their study also suggests that higher-order interactions can encode significant bio-makers that distinguish between healthy and pathological states associated with anesthesia or brain injuries, and they mirror changes associated with aging. Further studies are needed to clarify the role of higher-order interactions in the brain. We hope our inference method can provide a new tool and a novel perspective on this important open question in neuroscience.

## Discussion

The hypergraph reconstruction problem considered here is a special (but very interesting and important) case of data-driven equation discovery. Instead of inferring the full equations, we aim to determine which variables are linked (inferring the hyperedges). Because we only need to know the causal relationship among the variables, our algorithm is less sensitive to the choice of the nonlinear feature library (compared to, for example, SINDy). This allows us to work with high-dimensional and interconnected dynamical systems in a model-free fashion.

The main strength of THIS is that it can perform robust hypergraph inference with minimal prior knowledge of the underlying system—we only assume that it can be modeled as a set of coupled differential equations. Despite the system-agnostic nature of the method, we do have the ability to incorporate additional information about the system as it becomes available. For example, if we know that the underlying hypergraph is a simplicial complex, then we can utilize the downward inclusion condition to automatically infer all the lower-order interactions, circumventing one of the limitations of THIS. The same is true for hypergraphs for which no lower-order interactions overlap with higher-order interactions (i.e., anti-simplicial complexes and, to first approximation, random sparse hypergraphs). If one knows the coupling functions, then this information can be integrated into the sparse regression step to improve inference.

One limitation of THIS is that the sampling area for data needs to be well-balanced. Namely, the area should be large enough to adequately explore the system's dynamics (i.e., go beyond the linearizable region) but also not too large in order to preserve the accuracy of the Taylor approximation. As long as this balance is achieved, THIS is fully flexible in terms of where data came from in the state space. In our experiments on Kuramoto and Lorenz oscillators, we found that this

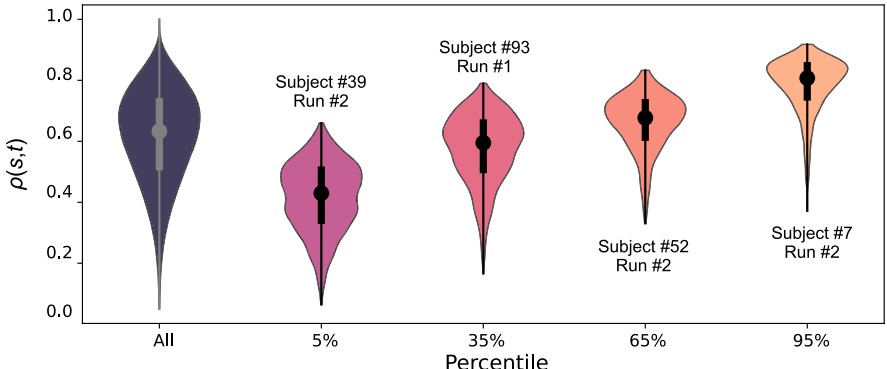
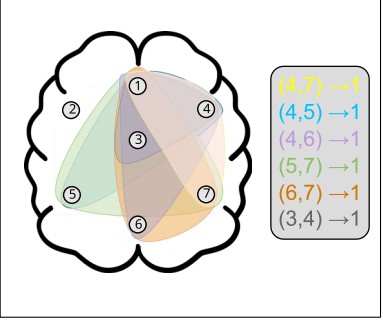

**Fig. 6 | Higher-order interactions play a significant role in shaping macroscopic brain dynamics.** Left panel: The percentage of dynamics (i.e., derivatives) contributed by the inferred nonpairwise couplings, $\rho(s, t)$. See Eq. (4) for the definition of $\rho(s, t)$. The left-most violin shows the distribution of $\rho(s, t)$ aggregated over all 218 time series $s$ and all time $t$. The five other violins show the distribution of $\rho(s, t)$ for one time series each. The time series displayed are the ones whose median ratio $\bar{\rho}(s)$ is at the 5, 35, 65, and 95 percentiles, respectively. Right panel: Illustration of the seven brain areas and the six most frequently inferred 3-edges. Interestingly, the top six hyperedges all point towards Area 1, which roughly corresponds to the prefrontal cortex (see Supplementary Fig. 7). This makes sense given that the prefrontal cortex is highly interconnected with the rest of the brain, known to be involved in a wide range of higher-order cognitive functions, and considered one of the key information processing hubs in the brain[72].

balance can be achieved for a wide range of sampling box sizes (Supplementary Fig. 2). A practical strategy for picking the right box size is the following: Instead of guessing one using intuition, we can test the algorithm over a wide range of box sizes and find the plateau for which the predictions stay stable. Moreover, we often have partial information about the connectivity in many applications. For instance, in the context of predicting missing links[85], the majority of the connections are known. We can utilize this information and tune the box size to maximize the match between the inferred connections and the known connections.

Another potential limitation of THIS, for some applications, is the computational complexity. In its current form, THIS can be applied to either a large number of nodes (about 100 nodes for pairwise and triadic interactions, see Fig. 3) or a wide range of interaction orders (see Supplementary Fig. 10 for an example of 64 nodes with interactions up to the fourth order), but not both at the same time. We suspect that any other inference algorithm would face the same difficulty, given the combinatorial complexity inherent to the problem (without a constraint on the maximum interaction order, one has to search over exponentially many potential interactions). Therefore, even though such numerical complexity is undesirable for an inference algorithm, it cannot be avoided without incorporating additional assumptions or constraints.

Nevertheless, we envision various workarounds to the above complexity limitation. First, in practice, we are often interested in inferring interactions up to a certain order. For any fixed interaction order $p$, we observe that the number of monomials to consider grows as $n^p$, which is much more tractable than combinatorial growth.

Moreover, to further reduce the computational cost, we can add a pre-processing step to filter out node pairs with low correlations and exclude them from the sparse regression calculation. This can effectively reduce the size of the monomial library. Figure 7 shows that applying this technique allows us to accurately reconstruct a 300-node random simplicial complex from generalized Kuramoto dynamics. We also show in Supplementary Fig. 3 that the computational cost of the algorithm increases with the system size as a power law $n^4$ for interactions up to the third order.

Other modifications could improve THIS by making it more robust against noise, taking advantage of recent advances in SINDy algorithms, most notably the Ensemble-SINDy[86] and Weak-SINDy[87] approaches. Finally, in many applications, one often has to work with partial measurements. In those cases, it would be important to

combine THIS with data assimilation techniques (e.g., ensemble Kalman filter[88]) to tackle missing variables.

We hope this work will further stimulate the development of hypergraph inference methods so that we can leverage the torrents of time-series data gathered in diverse fields to uncover previously hidden (higher-order) interactions in complex systems.

## Methods
### Implementation of SINDy
In THIS, SINDy is implemented with a library of $N$ monomials:

$$\mathcal{D} = \{1\} \cup \{x_i \mid i = 1, \ldots, n\} \cup \{x_i x_j \mid i, j = 1, \ldots, n\}$$
$$\cup \{x_i x_j x_k \mid i, j, k = 1, \ldots, n\} \cup \cdots$$

The purpose of SINDy is to find a parsimonious model that best explains the time-series data $X = \{\mathbf{x}(t_1), \mathbf{x}(t_2), \cdots, \mathbf{x}(t_K)\}$. The basic idea is to solve the matrix equations

$$\dot{\mathbf{x}}_i = \mathcal{D}(X)\mathbf{v}_i, \quad i = 1, \cdots, n \qquad (5)$$

with sparse, $N$-dimensional column vectors $\mathbf{v}_i$. The $n$ matrix equations (one for each node) are independent of each other and can be solved in parallel. Here, $\dot{\mathbf{x}}_i = [\dot{x}_i(t_1), \dot{x}_i(t_2), \cdots, \dot{x}_i(t_K)]^\top$ is a $K$-dimensional column vector consisting of the derivative of $x_i$ at different time points $t_k$ (either given as part of the data or inferred from $\mathbf{x}_i$). Each column of the $K \times N$ matrix $\mathcal{D}(X)$ corresponds to a monomial of the variables, whereas the rows encode different time points. In solving the matrix equation, we want to balance the goodness-of-fit (i.e., minimization of the mismatch between $\dot{\mathbf{x}}_i$ and $\mathcal{D}(X)\mathbf{v}_i$) with the sparsity of $\mathbf{v}_i$, which is achieved by performing sequential thresholded least square regression on Eq. (5). After solving the optimization problem, each nonzero element in $\mathbf{v}_i$ is linked to a monomial from the library, which can in turn be used to infer the existence of hyperedges pointing towards node $i$.

### Analysis of the EEG time series
We apply THIS to the resting-state time series from the dataset reported in refs. 89–91, consisting of 109 subjects, each with two independent recordings. Each recording lasts one minute, with a sampling rate of 160 points per second. For each inference, we start with a 64-dimensional signal obtained from 64 sensors distributed across the scalp. For the sake of tractability and noise reduction, we

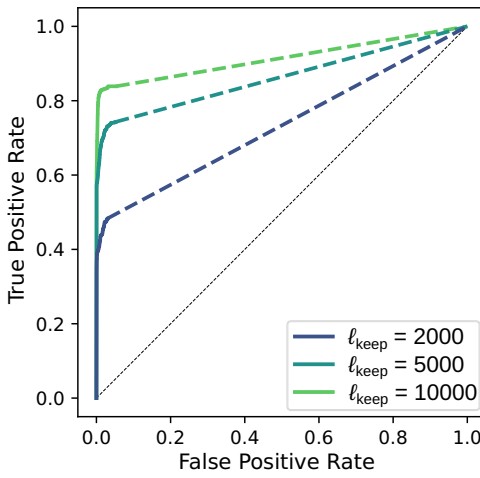

**Fig. 7 | A filtering strategy enables THIS to reconstruct even larger hypergraphs.** In order to reduce the size of the monomial library (thus saving computation and memory), we introduce a pre-processing step before inference to focus on the most relevant $\ell_{keep}$ node pairs. Here we choose the Pearson correlation coefficient as the filtering criterion because it is the standard measure in causal inference literature for excluding non-interacting nodes[93]. A plethora of other more advanced measures can be used to further refine the filtering step[94,95]. We show the ROC curves for the inference of a 300-node random simplicial complex from generalized Kuramoto dynamics at three different values of $\ell_{keep}$. Due to the exclusion of monomial terms, the ROC curves do not extend all the way to the upper right corner. Dashed lines indicate how the ROC curves would look like if we extend them by randomly adding hyperedges to the inferred structure. We see that not only is the filtering strategy able to significantly reduce computational cost without sacrificing accuracy, it also serves as a natural way to select the threshold $\epsilon$ (e.g., the end of the ROC curve for $\ell_{keep} = 10,000$ infers over 80% of the true positives with a near 0 false positive rate).

divided the sensors into seven groups according to their proximity and took the average within each group, reducing the 64-dimensional signal to a 7-dimensional signal (Supplementary Fig. 7). Each dimension captures the macroscopic dynamics of one of the seven brain regions. (We show in the Supplementary Fig. 10 that higher-order interactions continue to play an important role when we analyze the full 64-channel EEG data without grouping them into seven brain regions.) To facilitate approximating the time derivatives through finite differences (i.e., $\dot{x}_i(t_k) = (x_i(t_{k+1}) - x_i(t_k))/\Delta t$), we apply a low-pass filter to the 7-dimensional signal, keeping only part of the signal whose frequency is below $\approx 5$[Hz]. Additionally, we normalize the time series so their standard deviation is 1. This last step is important because having variances much larger or smaller than 1 induces large discrepancies in the order of magnitude for the monomials of different degrees, rendering SINDy's thresholding ill-conditioned. Finally, as THIS is valid in a restricted domain of the state space, we apply it to the 1000 data points that are closest to the median, which effectively tunes the sampling box size.

## Reporting summary
Further information on research design is available in the Nature Portfolio Reporting Summary linked to this article.

## Data availability
The EEG data used in this study are available in the PhysioNet database[89–91]. The data generated for this study have been deposited on `THIS`[92] repository: https://doi.org/10.5281/zenodo.10530470.

## Code availability
The codes created for this article have been deposited on `THIS`[92] repository: https://doi.org/10.5281/zenodo.10530470.

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

## Acknowledgements

We thank Federico Battiston, Maxime Lucas, Mason Porter, Andrew Stier, Marc Timme, and David Wolpert for insightful discussions. We are grateful to Benedetta Franceschiello and Giovanni Petri for their guidance in analyzing and interpreting the EEG data. R.D. was supported by the Swiss National Science Foundation under grants P400P2_194359 and 200021_215336. F.D. was supported by the National Centre of Competence in Research "Dependable, ubiquitous automation": https://nccr-automation.ch/. Y.Z. acknowledges support from the Omidyar Fellowship and the National Science Foundation under grant DMS-2436231.

## Author contributions

R.D., G.D.P., and Y.Z. designed the research, developed the methodology, and derived the theoretical results. R.D., G.D.P., F.D., and Y.Z. analyzed the numerical results and wrote the manuscript. R.D. wrote the code and ran the simulations.

## Competing interests

The authors declare no competing interests.
