## [Transparent Peer Review file · Nature Communications]

Hypergraph reconstruction from dynamics

Corresponding Author: Dr Yuanzhao Zhang

Version 0:

Reviewer comments:

Reviewer #2

(Remarks to the Author)

In the manuscript titled “Hypergraph reconstruction from dynamics”, the authors present a method for recovering higher-order structure from time-series data. This paper leverages the well-known SINDy algorithm by Taylor-expanding a system of ODEs about a point in the phase space and then interpreting “significant” monomial terms as multi-way interactions, (e.g., a monomial $x_i x_j x_k$ with a coefficient $a_{ijk} > \epsilon$ implies the existence of an interaction $\{i, j, k\}$). This method is applied to several small datasets with synthetic data generated from the Kuramoto model and empirical EEG data.

Overall, I find the method interesting and potentially useful, but some details remain unclear to me. The manuscript solely considers small networks and it is unclear whether this is due to computational reasons. In addition, some mathematical details should be expounded upon (details provided below).

Some additional comments and questions:

- * The manuscript states that ARNI is the only model-free approach. What about information-theoretic approaches? Some examples:
 - Luppi, A.I., Mediano, P.A.M., Rosas, F.E. et al. A synergistic core for human brain evolution and cognition. Nat Neurosci 25, 771–782 (2022). <https://doi.org/10.1038/s41593-022-01070-0>
 - Thomas F. Varley, Maria Pope, Maria Grazia Puxeddu, and Olaf Sporns. Partial entropy decomposition reveals higher-order information structures in human brain activity (Cited in the original manuscript)
 - Tomas Scagliarini, Daniele Marinazzo, Yike Guo, Sebastiano Stramaglia, and Fernando E. Rosas. Quantifying high-order interdependencies on individual patterns via the local O-information: Theory and applications to music analysis
- * On lines 104-105, adding “for all values of x ” would make it more precise.
- * Line 193: It would be helpful for EDMD to be written out. There are other acronyms that aren’t defined as well.
- * In Figure 1, more spaced-out $D(X)$ columns would be helpful. It would enhance the clarity to mention in the caption that the nodes/edges dashed/greyed out correspond to nodes/edges outside node i ’s neighborhood.
- * In several places, I thought “order” was a bit ambiguous. For example, on lines 86-87, the manuscript refers to the “ p -th order interaction function.” This would make sense when referring to polynomials, but in this instance, it seems as though the manuscript is trying to be more general than that. An alternative explanation could be that $h^{(p)}$ is an interaction function corresponding to a p -way interaction. In this case, however, a p -way interaction is defined as order $p-1$.
- * Line 146-149: What are the rare circumstances under which the coefficient can disappear?
- * How would the proposed framework be modified to capture directed relationships? As I understand it, the monomial basis chosen will yield the same coefficient for (i, j) and (j, i) . If not, the manuscript should describe this in further detail. If so, discussing how to generalize this approach to directed higher-order networks would be helpful.
- * It might be more helpful to state at the beginning of the “Benchmark on synthetic data” section that (1) for all examples, samples are drawn uniformly from an δ -box ($\prod_{i=1}^N [-\delta + x_i, \delta + x_i]$) and (2) for all examples that $\dot{\mathbf{x}}$ is calculated from the known equations. Then, for each example, you can state what \mathbf{x}_0 and δ are.
- * How well does the THIS framework perform for the synthetic data if you do not leverage knowledge of the equations? I.e., simulating the Kuramoto or Lorenz oscillators for a length of time. In that case, you cannot uniformly sample from a desired region and probably need to truncate the samples to lie within the δ -box. In this case, how does the time step affect the performance of THIS?
- * When Taylor-expanding around \mathbf{x}_0 , I’m not sure how one can guarantee that the higher-order Taylor terms do not mistakenly indicate that higher-order interactions are present. It would be helpful if the manuscript could talk more about this. For example, for the pairwise Kuramoto model, $\sin(x_i - x_j) = (x_i - x_j) + (x_i - x_j)^3/6 + \dots$. Will the SINDy

framework mistakenly find higher-order interactions? I understand that the ROC curve captures the sensitivity of the threshold, but it would be interesting to look at (1) only pairwise interactions and (2) only 3-interactions and see how well the inference performs.

* Minor point, but in “Implementation of SINDy” for the definition of \mathcal{D} , it could be helpful to (1) put the monomials of each order on a separate line and (2) end each monomial order with a monomial term instead of dots, to indicate that the monomials don't go up to arbitrary order. It could even be more compact to use set notation, e.g., $\mathcal{D} = \{1\} \cup \{x_i \mid i \in V\} \cup \{x_i x_j \mid i, j \in V\} \cup \{x_i x_j x_k \mid i, j, k \in V\}$.

* Is Figure 2 implying that ARNI is effectively a random classifier? Perhaps THIS should also be compared to an information-theoretic approach (described above).

* I think that the vertical axis of Fig. 4 should be relabeled to be more compact. Is there a way to more clearly visualize the hypergraph in Fig. 4? It's difficult to distill the message from the illustration.

* I appreciate the manuscript being accompanied by code for reproducibility.

(Remarks on code availability)

Reviewer #3

(Remarks to the Author)

The manuscript tackles the problem of hypergraph reconstruction from data using the Sparse Identification of Nonlinear Dynamics (SINDy) algorithm. While the topic is interesting and the paper is well-written, several critical limitations hinder the generality and impact of the results presented.

1. Algorithmic Advancement and Novelty: The manuscript essentially applies the SINDy algorithm, introduced in 2016, to the problem of identifying hyperedges. The entire hypergraph is treated as a unique monolithic nonlinear system, and SINDy is applied. While feasible, this approach represents an incremental rather than substantial advancement. Furthermore, the method significantly limits the size of hypergraphs that can be considered. The authors use a third-order approximation, including all monomials up to order three. Even with just two and three interactions, the number of functional bases (the monomials) increases dramatically with hypergraph size.

2. Scalability and Realism: The numerical validation examples involve very small hypernetworks (7 nodes), which is unrealistic for inferring hypergraph structures in natural or engineering systems. Typically, hypergraphs represent systems with a large number of interconnected subsystems, making the examples insufficient for practical applications.

3. Assumptions on Hypergraph Order: The paper assumes prior knowledge of the network's hypergraph nature. For instance, in the brain example, interactions up to order three are considered. It is unclear why interactions of order four or higher are not considered. Ideally, the algorithm should identify the appropriate interaction order within a certain range. When no prior knowledge is available, the algorithm should gradually increase the order of monomials, conduct the identification, and compare results with lower-order outputs. It should stop automatically once satisfactory identification is achieved based on defined metrics. The current approach exacerbates the curse of dimensionality problem as the order of monomials increases. Additional techniques are needed to handle this issue effectively.

4. Dimensionality and System Treatment: Hypergraphs are typically used to manage large interconnected systems by avoiding the need to treat the entire system as a single entity. However, the authors treat the hypergraph as a unique system, inheriting all the curse of dimensionality problems that hypergraph representations aim to mitigate.

5 Taylor Expansion and SINDy Application: The paper implies the use of a Taylor expansion for dynamic equations. In Figure 1, x_i and \dot{x}_i should be substituted with Δx_i and $\Delta \dot{x}_i$, respectively. For the Kuramoto and Lorenz examples, it seems the Taylor expansion is conducted about the origin, assuming the dynamics vanish at this point. More details on this aspect are needed.

6. Data-Driven Brain Hypergraph Reconstruction: For purely data-driven cases like brain hypergraph reconstruction, it is unclear how the authors address the $F_i(x_0)$ term. Do they consider differences in the data series for intervals when data are “close”? This point requires further elaboration.

7 Sparsity-Promoting Optimization: The authors mention a sparsity-promoting optimization approach but also use a threshold to neglect coefficients below a certain value. Clarification is needed on why this thresholding is necessary, and if so, how it should be selected. SINDy does not inherently involve an a posteriori coefficient regret procedure.

8: Brain Hypergraph Reconstruction Example:

Although interesting, the example on the brain hypergraph reconstruction suffers from severe scientific limitations. Firstly, it would be better to avoid grouping brain areas and consider a higher spatial resolution (otherwise an area could sum up aspects of many other surrounding areas limiting the informative power of the analysis). Secondly, and more importantly, the findings for the brain data analysis are not supported by clinical experiments nor any other form of validation. This makes the scientific findings questionable since, as also written by the authors, brain data are often subjected to diverse (and contradictory) interpretations. Although the findings might be of some interest in guiding further clinical/experimental validations, the authors should further reduce their claims on the provided data analysis.

In summary, this paper is clearly written and addresses an interesting topic. However, the issues outlined above prevent it from achieving the broad generality and novelty required for publication in Nature Communications. It may be more suitable for specialist journals or other journals focused on methodological advancements in network science and data-driven modeling.

(Remarks on code availability)

Reviewer #4

(Remarks to the Author)

I co-reviewed this manuscript with one of the reviewers who provided the listed reports. This is part of the Nature Communications initiative to facilitate training in peer review and to provide appropriate recognition for Early Career Researchers who co-review manuscripts

(Remarks on code availability)

Reviewer #5

(Remarks to the Author)

The authors propose a general method for inferring hypergraph structures from dynamic data. The paper is very well written and more importantly the inference method could potentially contribute to the long-lasting discussion in the community about whether the consideration of higher-order interactions is necessary in many real-world systems. I would like to recommend accepting the paper. In addition, I have two minor comments that I wish the authors could elaborate on.

1. In all the inference tasks, only two-body interactions and three-body interactions are considered. I have no problem with such a choice for simplicity, however, since the number of variables to infer increases polynomially as the order of interaction increases, is it possible to apply the method to (more) higher-order interactions with a reasonable computational cost?

2. Indeed there is the long-lasting discussion about whether higher-order interaction (and higher-order network models) are necessary to capture real-world dynamics. I believe an alternative question is that if a pairwise network model can provide a good enough approximation to real systems, such as brain networks. The authors examine the inference on real EEG data and calculate the contribution ratio between pairwise and three-body interactions to show the importance of higher-order interactions. As the authors have discussed in the first section, there are scenarios in which a higher-order interaction can be effectively decomposed as combinations of pairwise interactions and thus pairwise and higher-order interactions cannot be distinguished. Coming back to the EEG scenario, is it possible that the algorithm chooses one possible structure while there are equivalent (or at least approximately equivalent) structures that are formed exclusively by pairwise interactions? Is there an approach to examine this possibility, given that there is no ground truth available?

(Remarks on code availability)

Version 1:

Reviewer comments:

Reviewer #3

(Remarks to the Author)

I have carefully read the revised version of your paper and your responses to my comments as well as those of the other reviewers. I appreciate the effort you have put into addressing the concerns raised, and I acknowledge the improvements made.

However, I still have significant concerns regarding my initial comment on the scalability of the algorithm. Specifically, when I referred to the monolithic structure of the algorithm, I was pointing to the fact that it seems each node in the hypergraph needs to interact or relate with all other nodes, potentially involving all possible combinations of nodes up to a certain significant order. This, in my view, could introduce considerable computational challenges as the number of nodes increases.

Could you please clarify whether I am interpreting this correctly, or if there are aspects of the algorithm's design that mitigate this issue which I may have overlooked? Further elaboration on how scalability is addressed or managed would greatly help to clarify this concern.

Also, You mentioned that the algorithm can be executed on each node in parallel, which theoretically allows for a linear increase in computational capacity as the number of nodes grows. While this parallelizability is a positive feature, I remain

concerned that each node must still solve a problem with combinatorial complexity as the number of nodes increases. This suggests that even though you can distribute the computational load, the underlying complexity remains significant as the interactions between nodes grow.

For this reason, I respectfully disagree with your statement: "the more nodes there are, the more parallelizable our method will be." In fact, I believe that the combinatorial nature of the problem may make the method less scalable, not more, as the number of nodes increases. Additionally, the newly introduced condition (4) appears to further exacerbate the combinatorial complexity.

Could you provide further clarification on how you manage or mitigate this increasing complexity as the number of nodes grows? It would be helpful to see a more detailed explanation or any benchmarks that might support the claim of scalability.

I did not find an explicit clarification from the authors on this point, and it seems there may be some disagreement or a lack of consideration regarding the significance of this issue. I am still unsure whether I am misinterpreting this aspect, and I would appreciate more detailed clarification.

To put this into context with a real-world example: imagine a scenario where we have data from a social network. Since in principle there are clusters of mutual friends among the users, it is reasonable to assume that a hypergraph might exist in influencing, let's say, opinion formation dynamics on the network. Is it possible to verify such hypothesis with the proposed algorithm on a realistic social network with thousands of nodes? Is the algorithm able to identify possible higher order patterns for analysing the network or steering the users' opinion in such networks? Or are we restricted to some unrealistic case of one hundred (or say one thousand) people?

I believe that this aspect of scalability and generality is crucial for a paper submitted to Nature Communications. While I recognize that this may be one of the first model-free methods applied to hypergraphs, the paper should provide a comprehensive result rather than just a "first result." The novelty alone may not be enough without demonstrating broader applicability.

(Remarks on code availability)

Reviewer #4

(Remarks to the Author)

(Remarks on code availability)

Reviewer #5

(Remarks to the Author)

I apologize for the delay in providing a report. The authors have fully addressed my questions thus I recommend accepting the manuscript.

(Remarks on code availability)

Version 2:

Reviewer comments:

Reviewer #3

(Remarks to the Author)

I wish to thank the authors for considering my earlier review comments and the revisions they made to address my earlier concerns. While I appreciate the authors' effort particularly the new Figure S3 showing computational scaling, I still have several significant concerns that I believe the authors should address before providing a detailed answer to each of my comments below.

1. The computational scaling analysis needs to be properly commented in the main text by adding a paragraph explaining the observations. The authors only mention that Figure S3 shows a power law scaling but do not comment on how the computational time scales with both the system size and the interaction order. Note that the figure shows that such time becomes 10 times higher when the number nodes goes from approx 10 to 30... The authors claim this scales "gracefully" in their response but I believe a proper comment on how computational effort scales with N and p should be provided.

2. Your brain dynamics application is compelling and well suited to illustrate your method capabilities. However I remain concerned about its broader applicability. The example I suggested on opinion dynamics in realistic social network sizes highlights this. While you note the data availability challenges, I believe the more fundamental issue remains unanswered on whether the method could handle such systems even if the data were available. To address this issue:

3. you could more explicitly characterise the types and scales of systems where your method is more appropriate by explicitly adding into the revised manuscript a remark on this issue discussing the practical limitations of the method more directly. Also you could outline potential future algorithmic developments or extensions that might extend the method's reach and capability.

4. The statement about parallelizability has been clarified as referring to CPU utilisation rather than algorithmic scalability, which I appreciate. However, this reinforces my concern that the underlying computational complexity remains a significant challenge even with parallel execution,

I believe a more thorough and careful treatment of the limitations of the method and its advantages would strengthen the paper and provide a clearer guidance to potential users of the method.

(Remarks on code availability)

Reviewer #4

(Remarks to the Author)

(Remarks on code availability)

I co-reviewed this manuscript with one of the reviewers who provided the listed reports. This is part of the Nature Communications initiative to facilitate training in peer review and to provide appropriate recognition for Early Career Researchers who co-review manuscripts

Version 3:

Reviewer comments:

Reviewer #3

(Remarks to the Author)

Thank you for your revision efforts. However, my fundamental concerns about the scalability of the methods remain and were not addressed. I will leave to the editorial team to make a final decision about the suitability of this paper for Nature Communications.

(Remarks on code availability)

Reviewer #4

(Remarks to the Author)

(Remarks on code availability)

Reviewer #2 (Remarks to the Author):

In the manuscript titled “Hypergraph reconstruction from dynamics”, the authors present a method for recovering higher-order structure from time-series data. This paper leverages the well-known SINDy algorithm by Taylor-expanding a system of ODEs about a point in the phase space and then interpreting “significant” monomial terms as multi-way interactions, (e.g., a monomial $x_i x_j x_k$ with a coefficient $a_{ijk} > \epsilon$ implies the existence of an interaction $\{i, j, k\}$). This method is applied to several small datasets with synthetic data generated from the Kuramoto model and empirical EEG data.

Overall, I find the method interesting and potentially useful, but some details remain unclear to me. The manuscript solely considers small networks and it is unclear whether this is due to computational reasons.

We thank the reviewer for the constructive comments. We agree with the reviewer about the importance of considering larger systems. We have now made our algorithm more efficient by optimizing the generation of monomials. The improved algorithm can easily handle hypergraphs with 100 nodes (Fig. 3) as well as the full 64-channel EEG data (Fig. S9). In the Discussion section, we also propose and test a simple pre-processing strategy to exclude node pairs that are unlikely to be directly connected. This allows us to significantly reduce the number of monomials that need to be considered and further increase the number of nodes our algorithm can handle. We show the effectiveness of this strategy using a hypergraph of 300 Kuramoto oscillators (Fig. 7). We also want to stress that THIS utilizes the (sparse) least squares algorithm, which is one of the most scalable inference methods. If a (sparse) least squares approach doesn't scale, then it's unlikely that anything else will.

In addition, some mathematical details should be expounded upon (details provided below). Some additional comments and questions: The manuscript states that ARNI is the only model-free approach. What about information-theoretic approaches? Some examples:

- Luppi, A.I., Mediano, P.A.M., Rosas, F.E. et al. A synergistic core for human brain evolution and cognition. *Nat Neurosci* 25, 771–782 (2022). <https://doi.org/10.1038/s41593-022-01070-0>
- Thomas F. Varley, Maria Pope, Maria Grazia Puxeddu, and Olaf Sporns. Partial entropy decomposition reveals higher-order information structures in human brain activity (Cited in the original manuscript)
- Tomas Scagliarini, Daniele Marinazzo, Yike Guo, Sebastiano Stramaglia, and Fernando E. Rosas. Quantifying high-order interdependencies on individual patterns via the local O-information: Theory and applications to music analysis

Thank you for the very interesting references. These information-theoretic approaches mostly deal with generalizations of functional connectivity (correlated activities between different brain regions), which are different from the effective connectivity (causal influence between different brain regions) we are considering. We have now cited these references and discussed their relations to our work.

On lines 104-105, adding “for all values of x ” would make it more precise.

Thanks. We followed this suggestion. Indeed, this is more precise.

Line 193: It would be helpful for EDMD to be written out. There are other acronyms that aren't defined as well.

We have now defined all acronyms in the paper. We list the newly introduced acronyms here for completeness:

- Extended Dynamic Mode Decomposition (EDMD)
- intracranial encephalography (iEEG)
- functional Magnetic Resonance Imaging (fMRI)

In Figure 1, more spaced-out $D(X)$ columns would be helpful. It would enhance the clarity to mention in the caption that the nodes/edges dashed/greyed out correspond to nodes/edges outside node i 's neighborhood.

We have now modified Fig. 1 with more spaced out columns and updated the caption as suggested by mentioning that we highlight in color the couplings received by node i .

In several places, I thought “order” was a bit ambiguous. For example, on lines 86-87, the manuscript refers to the “ p -th order interaction function.” This would make sense when referring to polynomials, but in this instance, it seems as though the manuscript is trying to be more general than that. An alternative explanation could be that $h^{(p)}$ is an interaction function corresponding to a p -way interaction. In this case, however, a p -way interaction is defined as order $p - 1$.

We agree that there was some ambiguity in the definition of the “order” of an interaction and we had quite some discussions among ourselves to settle on a convention. What we try to make clear in the manuscript is that there is a p -th order interaction when a function $h^{(p)}(x_i, \dots, x_{i+p})$ involving p distinct variables is present in the dynamics. What makes this concept potentially confusing is that a p -th order interaction can be captured by a monomial of degree $p - 1$ in the Taylor series. For example, both the monomials $x_k^2 x_j$ and $x_k x_j$ could be associated with the Taylor expansion of $h^{(3)}(x_i, x_j, x_k)$ in the dynamics of node i and hence they are associated with the third-order interaction $(x_j, x_k) \rightarrow x_i$ even though the two monomials are of degree 3 and 2, respectively. We tried to clarify this aspect in the manuscript by commenting “Note that a monomial of degree $p - 1$ in the Taylor expansion corresponds to an interaction of order p .”

Line 146-149: What are the rare circumstances under which the coefficient can disappear?

The corresponding derivative of F_i evaluated at \mathbf{x}_0 could vanish, preventing the inference. For example, the derivative of x^2 at $x = 0$ vanishes, even though $2x$ is a nonzero function. We have now made this point more explicit in the text by saying “a Taylor coefficient could vanish when evaluated at \mathbf{x}_0 even when the corresponding interaction exists. We can easily circumvent this problem by choosing a different base point \mathbf{x}_0 .”

How would the proposed framework be modified to capture directed relationships? As I understand it, the monomial basis chosen will yield the same coefficient for (i, j) and (j, i) . If not, the manuscript should describe this in further detail. If so, discussing how to generalize this approach to directed higher-order networks would be helpful.

THIS identifies each node’s vector field, and thus its possible neighbours, independently. We do this node by node, therefore THIS captures directed interactions. In particular, THIS can distinguish the influence of the triplet $\{x_i, x_j, x_k\}$ on the dynamics of x_i from the influence of the same triplet on the dynamics of x_j . In this sense, THIS is sensitive to directed hyperedges as it identifies which node is being influenced and which nodes are the influencers. However, THIS cannot identify the ordering of the influencer nodes in the hyperedge, because $\partial_{j,k} F_i = \partial_{k,j} F_i$. Nevertheless, such a distinction does not really matter in our model-free setting. Indeed, we assume that we are completely ignorant regarding the interaction functions. Therefore, it does not really matter **how** the nodes influence each other, but rather **if** they influence each other.

It might be more helpful to state at the beginning of the “Benchmark on synthetic data” section that (1) for all examples, samples are drawn uniformly from an δ -box ($\prod_{i=1}^N [-\delta + x_i, \delta + x_i]$) and (2) for all examples that \mathbf{x} is calculated from the known equations. Then, for each example, you can state what \mathbf{x}_0 and δ are.

Thank you. We agree that clarifying these aspects enhances clarity. We have updated the manuscript accordingly, by specifying the value of δ and \mathbf{x}_0 and by explaining how the derivatives are measured.

How well does the THIS framework perform for the synthetic data if you do not leverage knowledge of the equations? I.e., simulating the Kuramoto or Lorenz oscillators for a length of time. In that case, you cannot uniformly sample from a desired region and probably need to truncate the samples to lie

within the δ -box. In this case, how does the time step affect the performance of THIS?

For the Lorenz oscillators, we already do what the reviewer suggested in the simulations (we use a single trajectory and the derivatives are estimated using finite differences). For Kuramoto oscillators, using fewer (but longer) trajectories, we achieve similar performance as in the original case. We show an example below, for which we take a time step of $h = 10^{-3}$ and time series of lengths $T \in \{5000, 6000, \dots, 10000\}$ (sampled from 9 to 16 continuous trajectories).

When Taylor-expanding around \mathbf{x}_0 , I'm not sure how one can guarantee that the higher-order Taylor terms do not mistakenly indicate that higher-order interactions are present. It would be helpful if the manuscript could talk more about this. For example, for the pairwise Kuramoto model, $\sin(x_i - x_j) = (x_i - x_j) + (x_i - x_j)^3/6 + \dots$. Will the SINDy framework mistakenly find higher-order interactions?

THIS would need a nonzero $\partial_{j,k}F_i$ term to infer the 3rd-order interaction $(x_j, x_k) \rightarrow x_i$. In the Reviewer's example, differentiation w.r.t. x_k would make the derivative function of $\sin(x_i - x_j)$ vanish.

I understand that the ROC curve captures the sensitivity of the threshold, but it would be interesting to look at (1) only pairwise interactions and (2) only 3-interactions and see how well the inference performs.

We performed the suggested analysis, please see Supplementary Figs. S3–S5.

Minor point, but in "Implementation of SINDy" for the definition of \mathcal{D} , it could be helpful to (1) put the monomials of each order on a separate line and (2) end each monomial order with a monomial term instead of dots, to indicate that the monomials don't go up to arbitrary order. It could even be more compact to use set notation, e.g., $\mathcal{D} = \{1\} \cup \{x_i \mid i \in V\} \cup \{x_i x_j \mid i, j \in V\} \cup \{x_i x_j x_k \mid i, j, k \in V\}$.

Thank you for the suggestion. We updated the "Implementation of SINDy" section as suggested.

Is Figure 2 implying that ARNI is effectively a random classifier? Perhaps THIS should also be compared to an information-theoretic approach (described above).

It would be a bit harsh to say that ARNI is completely random. What might lead to the poor performance in Fig. 2 is that ARNI detects interactions but cannot distinguish the interaction orders. The task THIS and ARNI perform is fundamentally different from what information-theoretic approaches tackle (effective connectivity vs functional connectivity, explained above).

I think that the vertical axis of Fig. 4 should be relabeled to be more compact. Is there a way to more clearly visualize the hypergraph in Fig. 4? It's difficult to distill the message from the illustration.

Thank you for the suggestion. We have made the label of Fig. 4 (Now Fig. 6) more compact and tried to improve the readability of the hypergraph.

I appreciate the manuscript being accompanied by code for reproducibility.

We thank the reviewer again for the constructive comments, which helped to significantly improve our manuscript.

Reviewer #3 (Remarks to the Author):

The manuscript tackles the problem of hypergraph reconstruction from data using the Sparse Identification of Nonlinear Dynamics (SINDy) algorithm. While the topic is interesting and the paper is well-written, several critical limitations hinder the generality and impact of the results presented.

We thank the reviewer for his/her detailed and constructive feedback.

Algorithmic Advancement and Novelty: The manuscript essentially applies the SINDy algorithm, introduced in 2016, to the problem of identifying hyperedges. The entire hypergraph is treated as a unique monolithic nonlinear system, and SINDy is applied. While feasible, this approach represents an incremental rather than substantial advancement.

We understand the reviewer's perspective. Inspired by the reviewer's comment, we further improved our algorithm (e.g., adding a filtering step to limit the size of the monomial library), which further distinguishes it from SINDy (more below). We would also like to argue that our algorithm represents a more than incremental advancement to the problem of hypergraph inference. First, dynamics on hypergraphs is an emerging new field and our algorithm is one of the first *model-free* methods that can reconstruct hypergraphs from time-series data. Second, we showed that THIS outperforms state-of-the-art competing methods such as ARNI. Third, our algorithm does not consider the hypergraph as a "monolithic" system: it can be parallelized and run independently for each node. Fourth, the performance of SINDy is very sensitive to the choice of the coordinate system and the nonlinear function library, which demands significant expertise from the user. Our algorithm, being properly adapted to the hypergraph inference problem, does not suffer from such limitations (you can apply the monomial library to almost any system). Finally, for inferring the relevant monomials we do not have to use SINDy. Other methods, such as EDMD, can also work.

Furthermore, the method significantly limits the size of hypergraphs that can be considered. The authors use a third-order approximation, including all monomials up to order three. Even with just two and three interactions, the number of functional bases (the monomials) increases dramatically with hypergraph size.

Thank you for pointing out this important issue. Motivated by the reviewer's comment, we made the algorithm more scalable. We are now able to reconstruct much larger hypergraphs. As illustrative examples, we applied the improved algorithm to synthetic systems consisting of 100 nodes (Fig. 3, Fig. S4, and Fig. S5) and to the 64-sensor EEG data directly without coarse-graining (Fig. S9). Moreover, in the Discussion section, we propose a simple filtering strategy to limit the size of the monomial library by discarding node pairs with low correlation. This allowed us to accurately reconstruct a random hypergraph with 300 nodes (Fig. 7).

Scalability and Realism: The numerical validation examples involve very small hypernetworks (7 nodes), which is unrealistic for inferring hypergraph structures in natural or engineering systems. Typically, hypergraphs represent systems with a large number of interconnected subsystems, making the examples insufficient for practical applications.

For many applications, the number of subsystems to be considered depends on the level of description. For different scientific questions, different granularity needs to be considered. As we showed with the EEG data, even with a small number of subsystems we can do something quite interesting! Understanding the effective connectivity between macroscopic brain regions is just as important as investigating more fine-grained brain networks. Moreover, we have now improved our algorithm so it can handle much larger systems (as discussed in detail above).

Assumptions on Hypergraph Order: The paper assumes prior knowledge of the network's hypergraph nature. For instance, in the brain example, interactions up to order three are considered. It is unclear why interactions of order four or higher are not considered. Ideally, the algorithm should identify the appropriate interaction order within a certain range. When no prior knowledge is available, the algorithm should gradually increase the order of monomials, conduct the identification, and compare results with lower-order outputs. It should stop automatically once satisfactory identification is achieved based on

defined metrics. The current approach exacerbates the curse of dimensionality problem as the order of monomials increases. Additional techniques are needed to handle this issue effectively.

This is a very good point. With our improved algorithm, we have been able to systematically explore higher-order interactions of all possible orders for the coarse-grained EEG data. We plotted the relative fitting error as a function of the number of interaction orders considered (Fig. 7). We find that including up to fourth-order interactions strikes a good balance between good reconstruction and parsimony. In Figs. S8 and S9, we show the relative contribution from the second-, third-, and fourth-order interactions to brain dynamics based on the coarse-grained EEG data and the full 64-channel EEG data.

Dimensionality and System Treatment: Hypergraphs are typically used to manage large interconnected systems by avoiding the need to treat the entire system as a single entity. However, the authors treat the hypergraph as a unique system, inheriting all the curse of dimensionality problems that hypergraph representations aim to mitigate.

We do not treat the hypergraph as a monolithic system. The algorithm can be run in parallel for each node—it identifies each node’s vector field independently. In order to stress this aspect we now mention in the text: “We would like to emphasize that our inference procedure can be done independently for each node. Thus, if one is only interested in a subset of nodes, there is no need to infer the full hypergraph. It also means that the more nodes there are, the more parallelizable our method will be, which is important for dealing with high-dimensional time series from large systems.”

Taylor Expansion and SINDy Application: The paper implies the use of a Taylor expansion for dynamic equations. In Figure 1, x_i and \dot{x}_i should be substituted with Δx_i and $\Delta \dot{x}_i$, respectively.

This is correct. In order not to overload Fig. 1 we inserted this clarification in the caption: “Note that, to be precise, the variables $\{x_i\}$ in $\mathcal{D}(\mathbf{X})$ should be interpreted as deviations from the Taylor-expansion base point \mathbf{x}_0 .”

For the Kuramoto and Lorenz examples, it seems the Taylor expansion is conducted about the origin, assuming the dynamics vanish at this point. More details on this aspect are needed.

For the Kuramoto example, we explicitly mentioned in the text: “Samples are taken randomly and uniformly inside a hypercube of side length $\delta = 0.1$ centered at the origin $x_0 = 0$. One can use hypercubes of side length up to $\delta = 1.0$ without affecting the results, see Supplementary Fig. S2.” For the Lorenz example, we specified: “The time series are all initialized within a hypercube of size length $\delta = 0.7$, centered at a random point x_0 chosen uniformly from $[-1, 1]^{3n}$ ”. Concerning the choice of the base point for the Taylor expansion, we wrote: “Moreover, the Taylor expansion can be computed around any point \mathbf{x}_0 where the vector field is differentiable, rendering the approach flexible in terms of where data are collected. In rare circumstances (usually with zero probability), a Taylor coefficient could vanish when evaluated at \mathbf{x}_0 even when the corresponding interaction exists. We can easily circumvent this problem by choosing a different base point \mathbf{x}_0 .”

Data-Driven Brain Hypergraph Reconstruction: For purely data-driven cases like brain hypergraph reconstruction, it is unclear how the authors address the $F_i(x_0)$ term. Do they consider differences in the data series for intervals when data are “close”? This point requires further elaboration.

The way in which the $F_i(x_0)$ term is addressed is via the constant term in SINDy, namely the first column in the $\mathcal{D}(\mathbf{X})$ matrix in Figure 1. In practice, we observed that the constant term can almost always be neglected.

Sparsity-Promoting Optimization: The authors mention a sparsity-promoting optimization approach but also use a threshold to neglect coefficients below a certain value. Clarification is needed on why this thresholding is necessary, and if so, how it should be selected. SINDy does not inherently involve an a posteriori coefficient regret procedure.

Having a separate threshold is not needed in principle, but it allows us to compute the ROC curve

a posteriori by varying the threshold from ∞ to 0 without the need to run sparse regression multiple times. We now mention in the text: “The threshold ϵ has overlapping functionalities with the sparsity parameter in SINDy. Having such a separate threshold facilitates measuring the quality of the inference—it allows us to draw the Receiver Operating Characteristic (ROC) curves a posteriori by adjusting ϵ without having to rerun the sparse regression algorithm multiple times.” Moreover, we now show in Fig. 5 that our main finding is not sensitive to the choice of ϵ .

Brain Hypergraph Reconstruction Example: Although interesting, the example on the brain hypergraph reconstruction suffers from severe scientific limitations. Firstly, it would be better to avoid grouping brain areas and consider a higher spatial resolution (otherwise an area could sum up aspects of many other surrounding areas limiting the informative power of the analysis).

As mentioned earlier, we believe that understanding the effective connectivity between macroscopic brain regions is just as important as investigating more fine-grained brain networks. That being said, our improved algorithm allowed us to analyze the 64-sensor EEG data directly without coarse-graining. This new analysis further confirmed our finding that higher-order interactions play an important role in shaping brain dynamics (Fig. S9).

Secondly, and more importantly, the findings for the brain data analysis are not supported by clinical experiments nor any other form of validation. This makes the scientific findings questionable since, as also written by the authors, brain data are often subjected to diverse (and contradictory) interpretations. Although the findings might be of some interest in guiding further clinical/experimental validations, the authors should further reduce their claims on the provided data analysis.

We agree with the reviewer that any interpretation of computational results concerning the human brain needs to be thoughtful and comes with caveats. For this purpose, we mention in the text: “We note that currently there is no consensus on the relative importance of pairwise versus nonpairwise interactions in shaping the macroscopic brain dynamics. Due to the intrinsic complexities of the brain and a lack of ground truth, different methods applied to different neurophysiological datasets can lead to seemingly contradictory results. ... Further studies are needed to clarify the role of higher-order interactions in the brain. We hope our inference method can provide a new tool and a novel perspective on this important open question in neuroscience.”

In summary, this paper is clearly written and addresses an interesting topic. However, the issues outlined above prevent it from achieving the broad generality and novelty required for publication in Nature Communications. It may be more suitable for specialist journals or other journals focused on methodological advancements in network science and data-driven modeling.

We thank the reviewer again for the constructive comments, which helped to significantly improve our manuscript.

Reviewer #4 (Remarks to the Author):

I co-reviewed this manuscript with one of the reviewers who provided the listed reports. This is part of the Nature Communications initiative to facilitate training in peer review and to provide appropriate recognition for Early Career Researchers who co-review manuscripts

We thank the Reviewer for taking the time to contribute to this review, which helped to strengthen our results.

Reviewer #5 (Remarks to the Author):

The authors propose a general method for inferring hypergraph structures from dynamic data. The paper is very well written and more importantly the inference method could potentially contribute to the long-lasting discussion in the community about whether the consideration of higher-order interactions is necessary in many real-world systems. I would like to recommend accepting the paper. In addition, I have two minor comments that I wish the authors could elaborate on.

We thank the Reviewer for their positive assessment of our work.

In all the inference tasks, only two-body interactions and three-body interactions are considered. I have no problem with such a choice for simplicity, however, since the number of variables to infer increases polynomially as the order of interaction increases, is it possible to apply the method to (more) higher-order interactions with a reasonable computational cost?

Thanks for your question. We addressed this point in the context of the brain example. We improved the efficiency of the algorithm so that we could perform an analysis of the 64-sensor EEG data directly without coarse-graining, looking at interactions up to fourth order (Fig. S9). Furthermore, for the coarse-grained EEG data with seven brain regions, we were able to look at higher-order interactions of all possible orders (Figs. S7 and S8). Also, in the Discussion section, we propose a pre-processing strategy to make our algorithm more scalable by discarding monomials associated with node pairs that show low correlation, which allows us to tackle even larger hypergraphs (Fig. 7).

Indeed there is the long-lasting discussion about whether higher-order interaction (and higher-order network models) are necessary to capture real-world dynamics. I believe an alternative question is that if a pairwise network model can provide a good enough approximation to real systems, such as brain networks. The authors examine the inference on real EEG data and calculate the contribution ratio between pairwise and three-body interactions to show the importance of higher-order interactions. As the authors have discussed in the first section, there are scenarios in which a higher-order interaction can be effectively decomposed as combinations of pairwise interactions and thus pairwise and higher-order interactions cannot be distinguished. Coming back to the EEG scenario, is it possible that the algorithm chooses one possible structure while there are equivalent (or at least approximately equivalent) structures that are formed exclusively by pairwise interactions? Is there an approach to examine this possibility, given that there is no ground truth available?

This is a very good question. To compare the two possibilities (pairwise and higher-order), we need to quantify how well the inferred hyperedges explain the observed dynamics. As pointed out by the Reviewer, our computation of the ratio between pairwise and nonpairwise contributions to the EEG dynamics is an attempt at such a quantification. In our setup, the inferred nonpairwise interactions cannot be decomposed into linear combinations of pairwise interactions, so there is no need to worry about equivalent structures formed exclusively by pairwise interactions. In other words, because we need higher-order monomials to adequately explain the EEG data, and such monomials can only come from genuine nonpairwise interactions, we can be confident that no pairwise networks can have the same effect.

Reviewer #3:

I have carefully read the revised version of your paper and your responses to my comments as well as those of the other reviewers. I appreciate the effort you have put into addressing the concerns raised, and I acknowledge the improvements made.

However, I still have significant concerns regarding my initial comment on the scalability of the algorithm. Specifically, when I referred to the monolithic structure of the algorithm, I was pointing to the fact that it seems each node in the hypergraph needs to interact or relate with all other nodes, potentially involving all possible combinations of nodes up to a certain significant order. This, in my view, could introduce considerable computational challenges as the number of nodes increases.

Could you please clarify whether I am interpreting this correctly, or if there are aspects of the algorithm's design that mitigate this issue which I may have overlooked? Further elaboration on how scalability is addressed or managed would greatly help to clarify this concern.

Also, You mentioned that the algorithm can be executed on each node in parallel, which theoretically allows for a linear increase in computational capacity as the number of nodes grows. While this parallelizability is a positive feature, I remain concerned that each node must still solve a problem with combinatorial complexity as the number of nodes increases. This suggests that even though you can distribute the computational load, the underlying complexity remains significant as the interactions between nodes grow.

For this reason, I respectfully disagree with your statement: "the more nodes there are, the more parallelizable our method will be." In fact, I believe that the combinatorial nature of the problem may make the method less scalable, not more, as the number of nodes increases. Additionally, the newly introduced condition (4) appears to further exacerbate the combinatorial complexity.

Could you provide further clarification on how you manage or mitigate this increasing complexity as the number of nodes grows? It would be helpful to see a more detailed explanation or any benchmarks that might support the claim of scalability.

I did not find an explicit clarification from the authors on this point, and it seems there may be some disagreement or a lack of consideration regarding the significance of this issue. I am still unsure whether I am misinterpreting this aspect, and I would appreciate more detailed clarification.

To put this into context with a real-world example: imagine a scenario where we have data from a social network. Since in principle there are clusters of mutual friends among the users, it is reasonable to assume that a hypergraph might exist in influencing, let's say, opinion formation dynamics on the network. Is it possible to verify such hypothesis with the proposed algorithm on a realistic social network with thousands of nodes? Is the algorithm able to identify possible higher order patterns for analysing the network or steering the users' opinion in such networks? Or are we restricted to some unrealistic case of one hundred (or say one thousand) people?

I believe that this aspect of scalability and generality is crucial for a paper submitted to Nature Communications. While I recognize that this may be one of the first model-free methods applied to hypergraphs, the paper should provide a comprehensive result rather than just a "first result." The novelty alone may not be enough without demonstrating broader applicability.

We thank the reviewer for the constructive comments. We agree with the reviewer that the scalability of the algorithm is an important challenge. Thus, we added a new figure explicitly showing how the computational cost of our algorithm increases with the system size (attached below and added to the manuscript as Supplementary Fig. S3).

Figure 1: Computation time of our algorithm as a function of the system size. We ran THIS on time series of length $T = 1000$ (blue) and $T = 2000$ (orange) for Kuramoto dynamics on sparse simplicial complexes with both second-order and third-order interactions. For each system size, the computation time is averaged over 5 different random simplicial complexes. A pre-filtering step was performed, keeping 10% of the most correlated pairs.

A few additional comments about scalability:

- Due to the nature of the hypergraph reconstruction problem, in principle, any algorithm would have to consider all other nodes in the system. Without prior knowledge, any node can be a potential neighbor to another node, this is an inherent aspect of the problem and thus a challenge for any algorithm (not just ours). In fact our algorithm still scales rather gracefully given this inherent complexity of the problem, as showcased in Figure 1.
- The reasoning above is one of the key motivations for leveraging prior knowledge and for our proposed pre-processing step (see Fig. 7 and the last paragraph of the Discussion Section). The basic idea is to exclude node pairs that are unlikely to be interacting from the inference calculations to improve scalability of the algorithm. This strategy works especially well for large sparse hypergraphs.
- In practice, we are often interested in inferring interactions up to a certain order. For any fixed interaction order a , the number of monomials to consider grows as N^a , which is much more manageable than combinatorial growth.
- As the reviewer noted, our algorithm is one of the first model-free methods applied to hypergraphs. With several hundred nodes, our method already opens new possibilities in investigating many important questions that were previously out of reach, which we illustrated using EEG data and brain dynamics in the manuscript.
- Regarding the real-world example proposed by the reviewer, it is indeed challenging to find time series data that captures the evolution of opinions in real-world networks, as such data often require high temporal granularity and detailed user activity logs, which are rarely available due to privacy concerns and the high cost of data collection. In our work, we emphasize the potential of our algorithm for applications in which high-quality time-series data are available, such as neuroscience.
- When we wrote that “the more nodes there are, the more parallelizable our method will be”, we simply meant that more nodes allow us to utilize more CPUs at the same time (one CPU performing inference for one node). This is not a comment about the scalability of the algorithm.

We thank the reviewer again for helping us to improve the manuscript.

Reviewer #3:

I wish to thank the authors for considering my earlier review comments and the revisions they made to address my earlier concerns. While I appreciate the authors' effort particularly the new Figure S3 showing computational scaling, I still have several significant concerns that I believe the authors should address before providing a detailed answer to each of my comments below.

1. The computational scaling analysis needs to be properly commented in the main text by adding a paragraph explaining the observations. The authors only mention that Figure S3 shows a power law scaling but do not comment on how the computational time scales with both the system size and the interaction order. Note that the figure shows that such time becomes 10 times higher when the number of nodes goes from approx 10 to 30... The authors claim this scales "gracefully" in their response but I believe a proper comment on how computational effort scales with N and p should be provided.

We have now dedicated a new paragraph in the Discussion section to the issue of computational cost and the power-law scaling in Figure S3. For example, we comment that "For any fixed interaction order p , the number of monomials to consider grows as N^p ."

2. Your brain dynamics application is compelling and well suited to illustrate your method capabilities. However I remain concerned about its broader applicability. The example I suggested on opinion dynamics in realistic social network sizes highlights this. While you note the data availability challenges, I believe the more fundamental issue remains unanswered on whether the method could handle such systems even if the data were available. To address this issue:

3. you could more explicitly characterise the types and scales of systems where your method is more appropriate by explicitly adding into the revised manuscript a remark on this issue discussing the practical limitations of the method more directly. Also you could outline potential future algorithmic developments or extensions that might extend the method's reach and capability.

Thank you for the suggestion. We now explicitly discuss the types and scales of systems where our method is more appropriate. For example, we comment that "In its current form, THIS can be applied to either a large number of nodes (about 100 nodes for pairwise and triadic interactions, see Fig. 3) or a wide range of interaction orders (see Fig. S10 for an example of 64 nodes with interactions up to the fourth order), but not both at the same time." We also outline potential future algorithmic developments or extensions that might extend the method's reach and capability. For example, see the discussion on pre-processing and other ideas towards the end of the main text.

4. The statement about parallelizability has been clarified as referring to CPU utilisation rather than algorithmic scalability, which I appreciate. However, this reinforces my concern that the underlying computational complexity remains a significant challenge even with parallel execution,

Yes, in its current form, our algorithm won't be able to infer fourth-order interactions for systems with thousands of nodes. However, we suspect that any other inference algorithm would face the same difficulty, given the combinatorial complexity inherent to the problem (without a constraint on the maximum interaction order, one has to search over exponentially many potential interactions). We have now emphasized this limitation. We would like to note, however, that even with this limitation, our method can already be applied to many interesting problems that were previously out of reach.

I believe a more thorough and careful treatment of the limitations of the method and its advantages would strengthen the paper and provide a clearer guidance to potential users of the method.

This is a great idea. We expanded our previous discussion on the strengths/limitations of the method in the Discussion section. Again, we thank the reviewer for helping us improve the manuscript.